# Krüppel-like Transcription Factor 7 Is a Causal Gene in Autism Development

**DOI:** 10.3390/ijms23063376

**Published:** 2022-03-21

**Authors:** Hui Tian, Shupei Qiao, Yufang Zhao, Xiyun Jin, Cao Wang, Ruiqi Wang, Yilin Wang, Yanwen Jiao, Ying Liu, Bosong Zhang, Jiaming Jin, Yue Chen, Qinghua Jiang, Weiming Tian

**Affiliations:** 1School of Life Science and Technology, Harbin Institute of Technology, Harbin 150080, China; hytianhui2021@163.com (H.T.); qiaoshupei43@163.com (S.Q.); vaejin_8811@163.com (X.J.); wc372935252@163.com (C.W.); 20b928039@stu.hit.edu.cn (R.W.); 1192800220@stu.hit.edu.cn (Y.W.); jiaoyw147@163.com (Y.J.); ly8363603@163.com (Y.L.); zbs96@foxmail.com (B.Z.); 20s028058@stu.hit.edu.cn (J.J.); yuechen1741@163.com (Y.C.); qhjiang@hit.edu.cn (Q.J.); 2Space Environment Simulation Research Infrastructure, Harbin Institute of Technology, Harbin 150080, China; zhaoyufang@hit.edu.cn

**Keywords:** ASD, klf7, regulatory gene, klf7^+/−^ mice, human brain organoids

## Abstract

Background: Autism spectrum disorder (ASD) is a complex neurodevelopmental disease. To date, more than 1000 genes have been shown to be associated with ASD, and only a few of these genes account for more than 1% of autism cases. Klf7 is an important transcription factor of cell proliferation and differentiation in the nervous system, but whether klf7 is involved in autism is unclear. Methods: We first performed ChIP-seq analysis of klf7 in N2A cells, then performed behavioral tests and RNA-seq in klf7^+/−^ mice, and finally restored mice with adeno-associated virus (AAV)-mediated overexpression of klf7 in klf7^+/−^ mice. Results: Klf7 targeted genes are enriched with ASD genes, and 631 ASD risk genes are also differentially expressed in klf7^+/−^ mice which exhibited the core symptoms of ASD. When klf7 levels were increased in the central nervous system (CNS) in klf7^+/−^ adult mice, deficits in social interaction, repetitive behavior and majority of dysregulated ASD genes were rescued in the adults, suggesting transcriptional regulation. Moreover, knockdown of klf7 in human brain organoids caused dysregulation of 517 ASD risk genes, 344 of which were shared with klf7^+/−^ mice, including some high-confidence ASD genes. Conclusions: Our findings highlight a klf7 regulation of ASD genes and provide new insights into the pathogenesis of ASD and promising targets for further research on mechanisms and treatments.

## 1. Introduction

Autism spectrum disorder (ASD) is a group of complex neurodevelopmental disorders. Over the past decade, the diagnostic rate of ASD has reached approximately 1.7% in the United States [1]. In Asia, there have been an increasing number of cases of ASD, and the incidence among men (0.45%) is higher than that among women (0.18%) [2]. ASD is a heterogeneous disorder with a heritability of 60–90% [3,4,5]. A few ASD cases are caused by single gene mutation, while most ASD cases are the accumulative effect of hundreds of small effects genes, known as complex idiopathic autism [6]. To date, more than 1000 genes have been shown to be associated with ASD [7], but only a few of these genes account for more than 1% of autism cases [8].

Many ASD risk genes are expressed preferentially in the developing cortex [9] and are inter-related at the transcriptional level [10], which indicates that transcriptional regulatory factors that are active during brain development may be associated with ASD. Klf7 is a mammalian zinc finger transcription factor (TF) whose family functions as activators, repressors, or both [11,12]. Klf7 is believed to regulate cell cycle progression and be involved in neurogenesis, cell proliferation and differentiation, promoting the growth of axons and maintaining the normal function of the nervous system [13,14]. Meanwhile, klf7 is also associated with the development of olfactory sensory neurons [15]. Loss of klf7 results in apoptosis of TrkA^+^ sensory neurons, showing decreased sensitivity to pain [16]. Klf7-null mice exhibit neonatal lethality and impaired axon projections in the olfactory system and visual system [17]. In addition, klf7 was proposed as a candidate gene for ASD, and emerging evidence has reported that patients with a deletion in 2q33.3q34, where the klf7 gene is located, exhibit ASD features [18,19,20,21]. These studies indicate a possible relationship between loss of klf7 and ASD. In addition, another study reported four unrelated individuals with de novo mutations in klf7 accompanied by ASD-related developmental delay/intellectual disability (DD/ID) and neuromuscular and psychiatric complications [22]. Owing to the transcriptional regulation activity and strong expression of klf7 in the central nervous systems [23], klf7 may be a causal gene in ASD.

## 2. Results

### 2.1. The Level of klf7 Is Altered in Human ASD Brain and klf7 Binds ASD Risk Genes

Single-cell RNA-sequencing (RNA-seq) data from ASD patient brains (*n* = 16 control, *n* = 15 ASD patients) [24] were used to determine klf7 levels. The clusters identified in the paper were grouped into more general cell types: oligodendroglia cells (ODC), projection neurons, microglia, excitatory neurons, inhibitory neurons, endothelial cell and astroglia. Klf7 expression was significantly decreased in ODCs (log2 (Fold Change) ‹log2FC› = −0.34968, adjusted *p* = 3.89 × 10^−8^), astroglia (log2FC = −0.42086, adjusted *p* = 1.09 × 10^−5^) and projection neurons (log2FC = −0.3741, adjusted *p* = 5.94 × 10^−11^) (Figure 1A–C), indicating that klf7 may be involved in ASD.

Many ASD risk genes are expressed preferentially in the developing cortex [9], while klf7 is required for the development of sensory neurons [25]. As a transcription factor, klf7 and ASD risk genes were initially expressed at the same stage, suggesting that there may be a link between klf7 and ASD genes. ChIP-seq of GFP-tagged klf7 in HEK293 cells identified 8692 target genes of human klf7 (GEO accession number: GSM2016749) [26], 264 of which were shared with the Simons Foundation Autism Research Initiative autism database (SFARI, Latest Release: 2021 Q1) [27,28] (Figure 1D). These ASD risk genes targeted by human klf7 were mainly enriched in processes related to long-term potentiation, circadian rhythm and synapse part (Figure 1E). To validate whether the ASD genes targeted by klf7 are inherent in mice, we mapped klf7 target peaks in the N2A cell line. Since ChIP-grade antibodies against klf7 are not available, N2A cells were transiently transfected with either HA empty vector or HA-klf7 expression vector. Western blotting with an anti-HA antibody was used to measure klf7 protein expression, which was consistent with klf7 mRNA expression (Appendix A). HA was enriched in immunoprecipitates in the IP group but not the IgG group (Appendix A). A total of 3418 reproducible peaks between two biological replicates were identified (Figure 1F). Klf7 preferentially bound to promoters (86.97%) (Appendix A), and the majority of peaks (81.73%) mapped within ±1 kb of the transcription start site (TSS) (Appendix A). Motif seeking validated that klf7 frequently binds high GC content sites (Appendix A), consistent with previous reports [17,29]. Gene ontology enrichment showed that all klf7 target genes were enriched in processes related to CNS development and neurons (Figure 1G). We compared klf7 target genes with those in the SFARI autism database and found 228 overlapping ASD risk genes (Figure 1H and Appendix A). These ASD target genes were evidently enriched in functions related to neurons and synapses (Figure 1I). Strikingly, klf7 transcriptionally regulates several high-confidence ASD risk genes, such as Pten, Shank1, Shank3, Auts2, Foxp1, and Pak2 (Appendix A). These findings raise the possibility that klf7 regulates other ASD genes during ASD development. In addition, klf7 regulates other transcription factors (TFs), such as sp1, Rest, Auts2, and Foxg1; thus, klf7 may control a transcriptional cascade network (Appendix A).

### 2.2. Effects of klf7 Deletion on the Adult Mouse

In order to test this hypothesis, we constructed Nestin-Cre conditional knockout mice (Appendix A). A significant decrease in klf7 expression was confirmed by qRT-PCR (*n* = 9) and Western blot (*n* = 4) (Appendix A). More than 90% of klf7^−/−^ mice died during the first two months after birth (WT (*n* = 30), klf7^+/−^ (*n* = 32), and klf7^−/−^ (*n* = 20)) (Appendix A). Laub et al. [17] found that klf7-null mice show little or no milk in the stomachs and severely hypoplastic olfactory bulbs (OBs) were the only overt anatomical abnormality, indicating that klf7 mice died from insufficient milk intake due to olfactory problems. In our study, we also found olfactory bulbs of klf7^−/−^ mice were shorter and smaller than those in WT mice (Appendix A). We hypothesized that the cause of death in klf7^−/−^ mice might be similar to that previously reported. Furthermore, some klf7^+/−^ mice and all klf7^−/−^ mice exhibited decreased body weight (WT (*n* = 29), klf7^+/−^ (*n* = 31), and klf7^−/−^ (*n* = 12)) and brain weight (WT (*n* = 24), klf7^+/−^ (*n* = 20), and klf7^−/−^ (*n* = 15)) (Appendix A). However, as long as they survived, the mice grew normally. The klf7^+/−^ mice first developed seizures at 4 months of age and peaked around 1 year old when they were slightly hit or squeezed (*n* = 30) (Appendix A). These data indicate that klf7 is necessary for the development of mice.

### 2.3. Klf7-Deficient Mice Show the Core Symptoms of ASD

Due to the high mortality rate of klf7^−/−^ mice, it was difficult to obtain enough population for all experiments. Thus, klf7^+/−^ mice were used. However, we characterized the behaviors of klf7^+/−^ mice and klf7^−/−^ mice aged 5–6 months to better understand the dosage effect on behavior (WT (*n* = 12), klf7^+/−^ (*n* = 12), and klf7^−/−^ (*n* = 7)).

In rodents, social interaction behaviors are closely related to the olfactory system [30]. A modified three-chamber experiment (Appendix A) and the food-burying experiment were designed. In modified three-chamber experiments, all mice spent more time in the food-containing chamber (Appendix A), interacted more with the food dish (Appendix A), displayed a shorter latency to find food (Appendix A) and entered the food-containing chamber more frequently (Appendix A). Although there was a significant difference in the number of entries between WT mice and klf7^−/−^ mice, we think this was due to avoidance of environmental changes, which is a characteristic of ASD [31]. All mice displayed a strong preference for food (Appendix A) during the food-burying test. Considering the normal responses to the stimulus, loss of klf7 did not significantly affect the olfactory system in surviving mice.

Subsequently, mice were subjected to a three-chamber sociability assay. WT, klf7^+/−^ and klf7^−/−^ mice showed no preference for either of the two empty cages during the habitation stage (Figure 2A and Appendix A). When we introduced a stranger mouse, the WT mice spent more time in the chamber containing mice (Figure 2B,F), and interacted more with the stranger mouse (Figure 2C). However, klf7^+/−^ mice spent less time in the stranger mouse-containing chamber (Figure 2B,F) and interacted with the stranger mouse for a shorter amount of time than the WT mice did (Figure 2C). Klf7^−/−^ mice spent more time in the empty chamber and did not interact with the stranger mouse (Appendix A). Next, another stranger mouse was placed. WT mice displayed a preference for and interacted frequently with the novel mouse, while klf7^+/−^ mice interacted with the novel mice less than WT mice (Figure 2D–F). Klf7^−/−^ mice showed no preference for interacting with the novel mice (Appendix A). The above results showed that male klf7-deficient mice displayed abnormal social interaction.

Self-grooming test and Y maze spontaneous selection test were performed to assess the repetitive behaviors. Klf7^+/−^ mice spent more time on grooming than WT mice (Figure 2G). Compared with WT mice and klf7^+/−^ mice, klf7^−/−^ mice showed the longest grooming time (Appendix A). In the Y maze spontaneous selection experiment, klf7-deficient mice preferred the original target arm while WT mice preferred to explore the novel arm (Figure 2H and Appendix A). Klf7^−/−^ mice exhibited a longtime jumping behavior (Appendix A), which is a previously reported stereotypical behavior [32].

### 2.4. Klf7-Deficient Mice Exhibit Other ASD-like Behaviors

Klf7-deficient mice exhibited hyperactivity in the open field test. Klf7-deficient mice traveled longer distances (Appendix A) and traveled faster (Appendix A). Klf7^−/−^ mice exhibited a decreased total resting time (Appendix A). In addition, klf7-deficient mice showed a poorer nest-building ability, which suggest a social interaction defect (Appendix A) [33,34]. Studies have shown that individuals with klf7 mutations are associated with intellectual disability [22]. In a novel object recognition experiment, both WT and klf7^+/−^ mice similarly recognized the novel object (Appendix A), while klf7^−/−^ mice showed a significantly decreased interaction time ratio (time spent interacting with the novel object/time spent interacting with the familiar object) (Appendix A) and a slightly decreased interaction frequency ratio (number of interactions with the novel object/number of interactions with the familiar object) (Appendix A). In the Morris water maze test, the klf7^+/−^ mice showed a decreased ability to learn on the fifth day and klf7^−/−^ mice showed obvious learning impairment during the five training periods (Appendix A). WT mice spent obviously more time in the target region and crossed the platform location more frequently than klf7^+/−^ mice. Klf7^−/−^ mice could not distinguish the target region at all (Appendix A). Klf7^+/−^ and klf7^−/−^ mice also had difficulty locating the platform (Appendix A). These data combined with the results of the novel object recognition test indicate that loss of klf7 impaired cognitive function and spatial learning and memory.

### 2.5. Klf7 Regulate a Large Number of Autism Genes

To investigate whether klf7 participates in ASD pathology by regulating other ASD risk genes, we compared the transcriptome of the whole brains of 1-month-old WT (*n* = 4) and klf7^+/−^ mice (*n* = 4) by RNA-seq. A total of 3609 differentially expressed genes (DEGs) (including 1643 upregulated and 1966 downregulated genes, log2FC > 1 or log2FC < −1, false discovery rate (FDR) < 0.01, Appendix A) were identified. Of the klf7 targeted ASD genes (a total of 228 genes), 43 showed significantly changed expression (with 23 showing upregulated expression and 20 showing downregulated expression, log2FC >1 or log2FC < −1, false discovery rate (FDR) < 0.01), and the other genes showed small changes (Figure 3A, −1 < log2FC < 1). Since klf7 also targets some TFs, to determine whether deletion of klf7 can indirectly cause other ASD genes to change, we compared the DEGs with the SFARI database (a total of 1004 genes). A total of 631 genes were shared with the SFARI database (Figure 3B and Appendix A); 215 genes of these genes were changed significantly (including 98 showing upregulated expression and 116 showing downregulated expression, log2FC > 1 or log2FC < −1, false discovery rate (FDR) < 0.01) and the remaining 417 genes showed small changes (Figure 3C, −1 < log2FC < 1, false discovery rate (FDR) < 0.01). Considering the large number of genes involved, the cumulative effect of their small changes cannot be ignored. We performed KEGG enrichment analysis and found that the DEGs were mainly enriched in processes and functions related to synapses, circadian entrainment and long-term potentiation (Figure 3D). We also focused our analyses on large-scale studies: of the 122 ASD risk genes identified in human midfetal deep cortical projection neurons by Willsey et al. [35], 69 were dysregulated in klf7^+/−^ mice (Appendix A, FDR < 0.01). The largest exon sequencing study on ASD to date (35,584 total samples; 11,986 samples from patients with autism) identified 102 risk genes [36], and 59 of these risk genes were affected in klf7^+/−^ mice (Appendix A, FDR < 0.01). This suggests that klf7 plays an important regulatory role on ASD genes. Thus, we compared the expression patterns of dysregulated ASD genes in klf7^+/−^ mice with that in ASD patients with significantly downregulated klf7 level [24]. A total of 103 genes in astrocytes (including 7 showing upregulated expression and 96 showing downregulated expression), 117 genes in ODCs (including 24 showing upregulated expression and 93 showing downregulated expression), and 264 genes in projection neurons (including 3 showing upregulated expression and 261 showing downregulated expression) exhibited a similar expression pattern (Appendix A), further suggesting that klf7 is involved in autism by regulating other autistic genes.

To ensure the high quality of RNA-seq data, we evaluated changes in the expression levels of a subset of high-confidence ASD risk genes, i.e., Shank3, Shank1, Cacna1b, Oxtr, Pcdh10 and Gad2, by qRT-PCR (*n* = 9) (Appendix A). qRT-PCR confirmed that klf7 can regulate some high-confidence ASD genes. Shank3 is deleted in all reported cases of Phelan–McDermid syndrome (PMS), a neurodevelopmental disorder that is characterized by autistic-like behaviors [37]. In addition, many Shank3 mutations have been identified in ASD patients with severe cognitive deficits who do not have PMS [38]. Pcdh10 mainly expressed in the basolateral amygdala, a brain region involved in social interaction of ASD. Male Pcdh10^+/−^ mice exhibit synaptic and behavioral deficits [39]. Oxytocin and oxytocin receptors (OXTR) play an important role in many social behaviors, and deficits in this system have been linked to ASD [40]. OXTR knockout mice exhibit decreased mother–offspring interaction [41] and impaired recognition [42]. Mutations in any one of these high-confidence ASD genes can lead to the development of ASD, and deletion of klf7 led to ASD genes which contain these high-confidence genes to be dysregulated. Thus, we proposed that the phenotype of klf7-deficient mice may be the cumulative or synergistic effects of multiple ASD genes in the transcriptional network.

### 2.6. Increasing the Expression of klf7 Rescued the Core Symptom of ASD in klf7^+/−^ Adult Mice

We also explored whether ASD symptoms can be rescued in adults. We intravenously administered 1 × 1011 vector genomes (vg) of adeno-associated virus (AAV, AAV-PHP.eB-GFP-control or AAV-PHP.eB-GFP-klf7) (*n* = 12 for each group). Four weeks later, we confirmed the transduction efficiency by assessing klf7 expression level. Compared with mice administered AAV-PHP.eB-GFP-control, increased klf7 was detected in both WT mice and klf7^+/−^ mice administered AAV-PHP.eB-GFP-klf7 (Appendix A).

Behavioral tests (*n* = 12 for each group) were performed to determine whether increasing klf7 in klf7^+/−^ adult mice can alleviate autism-like behavior. When we introduced the first stranger mouse in the three-chamber social interaction experiment, WT/AAV-control mice obviously spent more time in the stranger-containing chamber than empty chamber. Klf7^+/−^/AAV-control mice spent nearly equal amounts of time in each chamber; there was no significant difference. Klf7^+/−^/AAV-control mice spent significantly less time in the stranger-containing chamber than WT/AAV-control mice, suggesting impaired social ability. When we increased klf7 level in WT and klf7^+/−^ adult mice, WT/AAV-klf7 mice and klf7^+/−^/AAV-klf7 mice spent significantly more time in the stranger-containing chamber than the empty chamber. Though the total time in the klf7^+/−^/AAV-klf7 group was still significantly lower than that in the WT/AAV-klf7 group, it was significantly higher than the klf7^+/−^/AAV-control group and equal with that in the WT/AAV-control group. There was no significant difference between WT/AAV-klf7 and WT/AAV-control mice, although the time spent in the stranger-containing chamber increased in the WT/AAV-klf7 group. In addition, klf7^+/−^/AAV-control mice interacted with the stranger mouse for a shorter amount of time when WT/AAV-control mice obviously preferred to interact with the stranger mouse. The interaction time between klf7^+/−^/AAV-control mice and the stranger mouse was significantly less than that between WT/AAV-control mice and the stranger mouse. This impairment of social behavior was rescued in the treatment group, as the interaction time between the klf7^+/−^/AAV-klf7 mice and the stranger mouse was significantly increased, which was still significantly lower than that of WT/AAV-klf7 mice, but it was significantly higher than that of klf7^+/−^/AAV-control mice. Klf7^+/−^/AAV-klf7 mice showed normal behavior similar to that of WT/AAV-control mice (Figure 4A). When we introduced the second stranger mouse, klf7^+/−^/AAV-control mice did not show a preference for the stranger mouse-containing chamber. In contrast, WT/AAV-control mice apparently stayed in the stranger mouse-containing chamber rather than the familiar mouse-containing chamber. Klf7^+/−^/AAV-control mice spent significantly less time in the stranger mouse-containing chamber compared with WT/AAV-control mice. In the treatment group, both WT/AAV-klf7 mice and klf7^+/−^/AAV-klf7 mice significantly preferred to stay in the stranger mouse caged chamber. Although klf7^+/−^/AAV-klf7 mice did not spend significantly more time in the stranger mouse caged chamber compared with klf7^+/−^/AAV-control mice, the total time spent in the stranger mouse caged chamber by klf7^+/−^/AAV-klf7 mice was increased compared with that in the klf7^+/−^/AAV-control group. Moreover, klf7^+/−^/AAV-klf7 mice spent significantly more time in the stranger mouse-containing chamber than that in the familiar mouse-containing chamber. Though the klf7^+/−^/AAV-control mice spent significantly more time interacting with the stranger mouse, the duration of interaction was significantly shorter than that of WT/AAV-control mice. However, klf7^+/−^ mice treated with AAV-klf7 showed improved performance in terms of interaction time. Klf7^+/−^/AAV-klf7 mice showed a significant preference for interacting with strangers rather than familiar mice, and the duration of interaction with the stranger mouse was similar to that of WT/AAV-klf7 mice, with no significant difference. Moreover, klf7^+/−^/AAV-klf7 mice also spent significantly more time interacting with the stranger mouse compared with klf7^+/−^/AAV-control mice (Figure 4B). In the self-grooming test, AAV-control-treated klf7^+/−^ mice exhibited significant increased grooming time in contrast to WT/AAV-control mice. Elevating klf7 levels corrected this abnormal repetitive behavior significantly for klf7^+/−^ mice and did not affect WT mice. There was no significant difference between klf7^+/−^/AAV-klf7 mice and WT/AAV-control mice (Figure 4C).

We next examined the effect of AAV-klf7 on disturbed ASD risk genes in klf7^+/−^ mice. The expression of 427 of the 634 ASD risk genes was reversed (Figure 4D and Appendix A), and the expression levels of some ASD risk genes were significantly changed (Appendix A). These results further demonstrate that klf7 can regulate ASD genes in the nervous system.

### 2.7. ASD Risk Genes Were Dysregulated in a klf7 Knockdown Human Brain Organoid Model

The use of organoids is necessary because of the scarcity and unavailability of tissue from patients with neurodevelopmental disorders at many developmental stages [43,44]. Organoids not only simulate early brain development but also recapitulate the early postnatal stage after long periods of culture [45]. Early development of the ASD brain was modeled in vitro [46]. To further verify the role of klf7 in a human brain organoid model, we knocked down klf7 with a lentiviral short hairpin RNA (shRNA) 7 days after differentiation and cultured the organoids for 90 additional days. Klf7 expression in the knockdown group was reduced to 36.9% of that in the control shRNA group (Figure 5A (*n* = 3)). Deletion of klf7 caused a significant decrease in diameter (the diameter in the control group was 750.8 μm, and that in the klf7 knockdown group was 734.8 μm, *p* = 0.035; Figure 5B) (*n* = 20). We also performed RNA-seq analysis of the organoids at 90 days of differentiation. The RNA-seq data from klf7 knockdown organoids revealed that 517 ASD genes (adjusted *p* < 0.01) were shared between klf7 knockdown organoids and the SFARI autism database, and that 344 (adjusted *p* < 0.01) ASD genes were shared between klf7 knockdown organoids, klf7^+/−^ mice and the SFARI autism database (Figure 5C and Appendix A). Of the 517 dysregulated ASD risk genes in klf7 knockdown organoids, 496 genes showed prominent changes (including 134 that showed upregulated expression and 362 that showed downregulated expression, log2FC > 1 or log2FC < −1, adjusted *p* < 0.01), and the remaining 21 genes showed small changes (−1 < log2FC < 1, adjusted *p* < 0.01; Figure 5D,E). We also analyzed two other studies. The largest exome sequencing experiment in the ASD population to date identified 102 risk genes [36], 59 of which (including 14 that showed upregulated expression and 45 that showed downregulated expression, adjusted *p* < 0.01) were affected in klf7 knockdown organoids (Figure 5F). Of the 122 ASD risk genes identified in human midfetal deep cortical projection neurons by Willsey et al. [35], 58 genes (including 18 that showed upregulated expression and 40 that showed downregulated expression, adjusted *p* < 0.01) were dysregulated in klf7 knockdown organoids (Figure 5G). This suggests that klf7 plays an important regulatory role on high-confidence ASD genes, and loss of klf7 leads to transcription dysregulation of high-confidence ASD genes. Furthermore, we compared the expression patterns of abnormal ASD genes in klf7 knockdown organoids with that in ASD patients with significantly downregulated klf7 expression [24]. A total of 346 genes in astrocytes (including 26 showing upregulated expression and 320 showing downregulated expression), 317 genes in ODCs (including 38 showing upregulated expression and 279 showing downregulated expression), and 370 genes in projection neurons (including 4 showing upregulated expression and 366 showing downregulated expression) showed similar expression patterns (Figure 5H–J). These observations verify that klf7 is involved in autism by mediating dysregulation of other ASD genes in the human brain, indicating that klf7 is a promising target for further research on mechanisms and treatments.

## 3. Discussion

Klf7 has been proposed as a candidate ASD gene, and loss of klf7 causes autistic features in patients [19,20]. Klf7-deficient mice showed the core symptoms of ASD and could not distinguish the target quadrant in the water maze experiment, suggesting that klf7-deficient mice had impaired spatial learning ability and memory, which is consistent with the cognitive delay and intellectual disability seen in patients with klf7 mutations [22]. Patients with loss of klf7 show complex partial seizures [18], which is consistent with the epilepsy-related characteristics of klf7^+/−^ mice. In addition, klf7-deficient mice exhibited low birth weight, which is observed in patients with 2q33.3q34 deletion [21]. Our klf7^+/−^ model mice were able to replicate the core symptoms of ASD and exhibit other accompanying symptoms of ASD similar to those of patients with klf7 mutations. Therefore, klf7^+/−^ mice can function as a model for studying the pathogenesis of ASD.

There are a thousand ASD susceptibility genes, and each gene has small disease-causing power [46]. We showed that klf7 is a causal gene in ASD, and that the dysregulated genes in klf7^+/−^ mice and the human brain organoids model are enriched for ASD risk genes. Rather than causing significant changes in the expression of a few specific genes, klf7 deficiency led to small global changes in the expression levels of 419 ASD risk genes similar to those observed in the brains of people with ASD and CHD8^+/−^ mice [47]. Researchers agree that the traits of ASD are determined by the dose effects of multiple genes [48,49]; thus, the cumulative effects of changes in many ASD risk genes should not be ignored.

Currently, reported ASD-related regulatory genes can be divided into two categories. One category is those that encode RNA-binding proteins, which are involved in the post-transcriptional regulation. Examples of these genes include FMRP [50], CPEB4 [51] and CELF4 [52]. The other category of regulatory genes encodes DNA-binding proteins, which are involved in regulating gene expression. TBR1 binds 23 ASD risk genes [53], while CHD8 binds and regulates 47 ASD genes [54]. To our knowledge, klf7 is the TF that can regulate the largest number of ASD risk genes at present. In addition to binding directly to ASD risk genes, klf7 regulates other ASD risk genes, such as Gad1, Gad2, CNTNAP3 and Pcdh10 in trans. To our surprise, klf7 deletion affected other regulatory genes. Loss of klf7 led to a 0.43-fold increase in the CPEB4 expression level, which is consistent with the slight increase in transcript levels in the brains of idiopathic ASD patients [51]. In addition, ChIP-seq data of klf7 in HEK293 cells showed that klf7 can target CPEB4. It may be that klf7 can regulate ASD genes at the post-transcriptional level through CPEB4, and this further illustrates the critical role of klf7 in regulating ASD genes. However, klf7 was not highlighted in previous studies, possibly because it shows a slight change in ASD individuals [24]. Considering the broad spectrum of genes it regulates, small changes in klf7 expression should be considered.

In another ASD model of maternal immune activation (MIA), we found that klf7 was upregulated at E17.5 and downregulated at P0 [55]. Epidemiological evidence is growing that environmental exposure early in development has an impact on ASD susceptibility later in life [56]. To support this theory, growing evidence from animal studies indicate that prenatal and early postpartum environmental factors can lead to changes in epigenetic programming and subsequent changes in disease risk [57]. Some environmental effects appear to be passed on through future generations. In utero exposure to MIA leads to a downregulated klf7 level in mice at P0, suggesting that there may be a link between environmental exposure, altered klf7 level and the development of autism. This is an interesting hypothesis and the next step in our research.

ASD patients usually display symptoms before the age of 3 [58], at which point brain development is almost complete. Another key question is whether this pathology can be reversed in adults. Currently, rescue methods used in ASD models mainly include those that maintain the balance between GABAergic and glutamatergic transmission [34], modify abnormal signaling pathways [59] and induce oxytocin release [60]. Klf7 overexpression can promote axonal regeneration and leads to functional recovery after nerve injury [61,62,63]. When we increased the level of klf7 in the brain of adult klf7^+/−^ mice, we found that it not only alleviated the core symptoms of ASD but also reversed the abnormal expression level of a majority of ASD genes. Whether exogenous enhancement of klf7 levels leads to other problems remains to be studied in the future, but the current results confirm that klf7 may be a target for the treatment of ASD.

## 4. Materials and Methods

### 4.1. Mice

Klf7 knockout mice were purchased from Cyagen Biosciences Inc. (Suzhou, China) and generated by crossing mice expressing a klf7 allele in which exon 2 was deleted with WT mice on the C57BL/6J background. Mice were housed 4 per cage, and food and water were freely available. The mice were kept in a temperature-controlled environment on a 12/12 h light–dark cycle, with lights on at 08:00. All animal husbandry and experimental procedures were approved by the Experimental Animal Welfare Ethics Committee of Harbin Institute of Technology (IACUC-2020036). Male mice were used for behavioral analysis in this study, and researchers were blinded to the genotypes of the mice.

### 4.2. Differential Gene Expression Analysis of Single-Cell Data from ASD Patient Brains

To explore the expression changes of klf7 in human ASD patients, we downloaded scRNA-seq data of ASD patient brains from UCSC Cluster Browser (https://autism.cells.ucsc.edu, accessed on 2020 November 20). Cell types were grouped into seven categories, including oligodendroglial cells (ODC), projection neurons, microglia, excitatory neurons, inhibitory neurons, endothelial cell and astroglia. A t test was used to identify DEGs between ASD and controls with a threshold of |log2FC| > 0 and *p* < 0.05.

### 4.3. ChIP-Seq

ChIP-seq was performed essentially as described previously [64]. The cells were crosslinked with 1% formaldehyde for 10 min and neutralized with glycine (150 mM final). After ultrasonic treatment, the cells were lysed and chromatin was cleaved into about 300 bp fragments. ChIP-grade anti-HA antibody (Abcam, ab9110, Cambridge, UK) and isotype control antibody (Abcam, ab172730) were used for immunoprecipitation overnight at 4 °C, and the samples were incubated with magnetic beads. The bound proteins were eluted from the beads and reversed crosslinking overnight at 65 °C. DNA was extracted with Phenol-chloroform for sequencing. The purified DNA was used to prepare for ChIP-seq library. Subsequently, the samples were pair-end sequenced on Illumina platform (Illumina, CA, USA). Library quality was assessed using the Agilent Bioanalyzer 2100 system.

Raw data (raw reads) in fastq format were firstly processed using fastp (version 0.19.11) software. In this step, clean data (clean reads) can be obtained by removing reads that contain adapters, reads that contain ploy-N and reads of low quality from raw data. Meanwhile, Q20, Q30 and GC content of clean data were calculated. All the downstream analyses were based on high-quality clean data. Clean reads were aligned to the mouse reference genome (Ensemble_GRCm38.94) using BWA mem (v 0.7.12). After mapping reads to the reference genome, MACS2 (version 2.1.0) peak calling software was used to identify IP enrichment regions on the background. A q-value threshold of all data sets was 0.05. After peak calling, chromosome distribution, fold enrichment, peak width, significant level and peak summit number were displayed. The de novo sequence motif and the matched known motifs were detected using Homer. Reproducibility was defined as overlapping peaks between two biological ChIP-seq replicates. GO enrichment analysis was implemented by the GOseq R package and KOBAS software was used to test the statistical enrichment of peak-related genes in KEGG pathways.

### 4.4. Behavioral Tests

All behavioral tests were performed during the light phase of the cycle between 09:00 and 17:00. All the mice for behavioral tests were allowed 2 h to habituate to the testing room before the experiment. The number of mice in WT and klf7^+/−^ group was twelve and the number of mice in klf7^−/−^ group was seven. When the treated mice were processed for behavior tests, the number of mice in each group was twelve. The experimenters were blinded to the genotypes of the mice during the testing phase.

#### 4.4.1. Olfactory Discrimination Test

The olfactory discrimination ability of the mice was evaluated using a modified three-chamber test [33]. Tightly sealed dishes containing food served as stimuli. During the habituation phase, a tightly sealed empty dish with holes was placed in the left and in the right chamber. The test mouse was placed in the middle chamber and allowed to freely explore the apparatus for 10 min. During the test phase, a dish containing holes that allowed the release food odors was placed in one chamber, and an empty dish was placed in another chamber, and the mice were allowed to explore the apparatus for 10 min. The time spent in each chamber, time spent within the area 5 cm around the dish, the number of entries into each chamber, and the latency to find the food for the first time were recorded. The three-chamber apparatus was cleaned with 75% alcohol and wiped with paper towels at the end of each experiment. Additionally, a food-burying experiment was also used to assess olfactory ability. Clean bedding was placed in a cage, and a piece of food was buried in the bedding. The latency to find the food for the first time was recorded. All data are presented as the means ± SEMs and were analyzed using one-way ANOVA.

#### 4.4.2. Three-Chamber Test

The test was performed as described previously [32]. The three-chamber apparatus consisted of a left, center and right chamber (25 × 16.7 cm) separated by two partitions. There was a square opening (5 × 5 cm) in the bottom center of each partition. A stranger mouse was placed in a wire cage. The three-chamber apparatus and wire cages were cleaned with 75% alcohol and wiped with paper towels at the end of each experiment. In the first 10 min session, the test mouse was placed in the center chamber, and the left and right chambers each contained an empty wire cage. The test mouse was allowed to freely explore the three-chamber apparatus. In the second 10 min session, an age- and sex-matched mouse (M1) to which the test mouse had never been exposed was placed in a wire cage, while the other wire cage was left empty (E). The test mouse was placed in the center chamber and allowed to freely explore the apparatus for 10 min before being removed. In the final 10 min session, a second age- and sex-matched mouse (M2) to which the test mouse had never been exposed was placed in the empty wire cage (E). Thus, the test mouse had to choose between interacting with the familiar mouse (M1) and interacting with the new stranger mouse (M2). The test mouse was placed in the center chamber and allowed to freely explore the apparatus for another 10 min. The time spent in each chamber and within a 5 cm radius of each wire cage were analyzed. All data are presented as the means ± SEMs and were analyzed using two-way ANOVA.

#### 4.4.3. Grooming

Repetitive behaviors were observed based on method with minor modifications [65]. The mice were caged singly for a 10 min accommodation period, and the time spent grooming within the subsequent 10 min was measured. All data are presented as the means ± SEMs and were analyzed using one-way ANOVA.

#### 4.4.4. Novel Object Recognition Test

The apparatus used for the novel object recognition test was a rectangular cage (25 × 50 cm) covered with fresh bedding. Each cage was used for only one mouse. During the test phase, the mice were placed in a cage and allowed to freely explore for 10 min. Next, two objects of the same size and color were placed in the cage. The test mouse was then placed in the middle of the arena and allowed to freely explore the cage, including two new objects, for 10 min. Twenty-four hours later, one of the objects was replaced with another novel object of a similar size but different shape and color. The same test mouse was then placed in the center and allowed to explore the cage and two objects. The ratio of the time spent interacting with the novel object to the time spent interacting with the familiar object and the ratio of the number of interactions with the novel object to the number of interactions with the familiar object were analyzed. All data are presented as the means ± SEMs and were analyzed by unpaired t test.

#### 4.4.5. Y Maze Spontaneous Selection Experiment

The forced alternation task was performed with modifications [66]. The Y maze apparatus comprised three identical arms (30 × 20 × 10) with a removable door at the entrance of each arm. The test mouse was placed in the central arm, and the mouse was allowed to choose to explore the left or right arm. Then, the door of the arm chosen by the mouse was removed for five minutes. Next, the mouse was removed and placed in the middle arm. The number of times that the mouse explored the originally selected arm and the novel arm over 10 trials was recorded. All data are presented as the means ± SEMs and were analyzed using one-way ANOVA.

#### 4.4.6. Open Field Test

The open field test was performed as described in previous reports [47]. The open field apparatus was a 40 × 40 × 30 cm box. A camera was used to record the activity of the mice. The test mouse was placed in the apparatus for 10 min and allowed to adapt to the environment. The activity of the mouse in the subsequent 10 min was quantified. The data are shown as the means ± SEMs and were analyzed by one-way ANOVA.

#### 4.4.7. Morris Water Maze Test

The Morris water maze was adopted to assess spatial learning ability and memory. The test was performed as previously described with few modifications. Each mouse was subjected to 4 trials a day for 5 days. The latency to locate the escape platform was recorded. After the training trials, the mice underwent a probe trial. In the probe trial, the platform was removed, and each test animal was given 60 s to search for the platform. The time spent in each quadrant and the number of platform location crossings (platform crossings) were recorded. The data are shown as the means ± SEMs and were analyzed by two-way ANOVA and one-way ANOVA.

#### 4.4.8. Nest-Building Test

A mouse was placed individually in a new cage containing 5 × 5 cm of cotton in a random corner. After 12 h, the constructed nest was scored on a scale of 1–5 according to the amount of cotton used and the height of the nest as follows [67]: (1) the cotton had not been touched; (2) the cotton was partially torn; (3) most of the cotton had been torn apart, but there was no sign of a complete nest; (4) a recognizable but flat nest; and (5) a perfect nest. All data are shown as the means ± SEMs and were analyzed using one-way ANOVA.

### 4.5. RNA-Seq and Analysis

Sample preparation and RNA extraction were performed as described previously [64]. Total RNA was extracted from the whole brains of 1-month-old WT and klf7^+/−^ mice associated with Figure 3, whole brain of WT and klf7^+/−^ mice 4 weeks after injected with AAV-NC virus and AAV-klf7 virus, and klf7 knockdown organoids derived from hiPSCs with a TRIzol plus total RNA extraction kit (Solarbio R1200). Specifically, whole brains from 4 mice for each genotype and 20 organoids for each group were pooled as one biological repeat. Total amounts and integrity of RNA were assessed using the RNA Nano 6000 Assay Kit of the Bioanalyzer 2100 system (Agilent Technologies, Santa Clara, CA, USA).

Next was library preparation, read mapping, and quantification of gene expression level. After the library is qualified, the different libraries are pooled according to the effective concentration and the target amount of data off the machine, then sequenced by the Illumina NovaSeq 6000. The end reading of 150 bp pairing is generated. Reads were aligned to the human genome (homo_sapiens_grch38_p12) and the mouse genome (mus_musculus_grcm38_p6) using Hisat2 (v2.0.5). The featureCounts v1.5.0-p3 was used to count the read numbers mapped to each gene. Then, FPKM of each gene was calculated based on the length of the gene and reads count mapped to this gene.

Next was the differential expression analysis and enrichment analysis. For DESeq2 with biological replicates, the DESeq2 R package (1.20.0) was used to analyze differential expression gene of two conditions/groups (two biological replicates per condition). To control the false discovery rate, Benjamini and Hochberg’s approach was used to adjust *p*-values. Padj < 0.01 and |log2(foldchange)| > 1 were set as the threshold for significantly differential expression. For edgeR without biological replicates, prior to differential gene expression analysis, for each sequenced library, the read counts were adjusted by edgeR program package through one scaling normalized factor. Differential expression analysis of two conditions was performed using the edgeR R package (3.24.3). The *p*-values were adjusted using the Benjamini and Hochberg method. Padj < 0.01 and |log2(foldchange)| > 1 were set as the threshold for significantly differential expression. We used clusterProfiler R package (3.8.1) to test the statistical enrichment of differential expression genes in KEGG pathways and gene ontology (GO) enrichment.

### 4.6. Intravenous Administration

We purchased a recombinant AAV (PHP.eB) expressing mouse klf7. The vector (CMV bGlobin-MCS-EGFP-3FLAG-WPRE-hGH polyA) expressed EGFP, which allowed determination of expression efficiency. A total of 1 × 1011 vg of AAV-PHP.eB was intravenously injected into male mice aged 3 to 5 months. Behavioral analysis, RNA-seq, and immunofluorescence were performed 4 weeks after injection, and the detailed experimental methods are described above.

### 4.7. shRNA-Mediated klf7 Knockdown

Organoids derived from hiPSCs (cell name: DYR0100, serial no.: SCSP-1301, National Collection of Authenticated Cell Cultures) were obtained from Zhejiang Hopstem Biotechnology Co., Ltd. (Hangzhou, China). After 3 days of differentiation, they were cultured in a 5% CO_2_ incubator at 37 °C. Half of the medium (Cat # HopCell-3DM-001a) was changed 2–3 times a week. On the 7th day of differentiation, 20 μL (1.0 × 10^5^ infectious units of virus) of either shklf7 or control shRNA lentivirus was added to each well (20 samples for each group), and the infected organoids were further cultured for 90 days before being harvested for RNA-seq.

### 4.8. Statistical Analysis

All data are shown as the mean ± SEM and were analyzed using unpaired t test, one-way ANOVA with Tukey’s post hoc comparisons test, or two-way ANOVA with Bonferroni’s post hoc comparisons test. All statistical analyses were performed using GraphPad Prism 5.

## Figures and Tables

**Figure 1 ijms-23-03376-f001:**
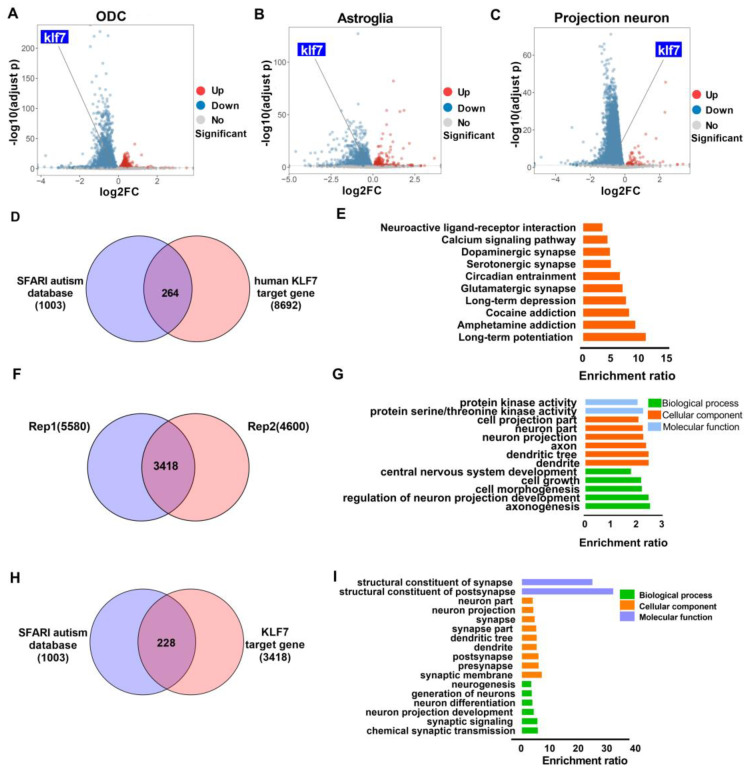
Klf7 targeted genes are enriched for ASD risk genes. (**A**–**C**) The klf7 level in the prefrontal cortex (PFC) of ASD patients according to human brain single-cell RNA-seq (*n* = 16 control, *n* = 15 ASD patients). Klf7 expression decreased significantly in ODCs (log2FC = −0.34968, adjusted *p* = 3.89 × 10^−8^), astroglia (log2FC = −0.42086, adjusted *p* = 1.09 × 10^−5^) and projection neurons (log2FC = −0.3741, adjust *p* = 5.94 × 10^−11^). (**D**) Shared genes between human klf7 target genes and SFARI autism database. The number of ASD risk genes from SFARI database bound by klf7 is shown, and the total number in each subset is presented in parentheses. (**E**) Functional enrichment analysis of ASD risk genes targeted by human klf7. These ASD risk genes were mainly enriched in processes related to long-term potentiation, circadian rhythm and synapse part. (**F**) Klf7-binding sites in two biological replicates of N2A cells relative to the IgG control. The peak number of biological replicates is shown in the Venn diagram. (**G**) Functional annotation of all klf7 target genes. All genes were enriched in processes related to CNS development and neurons. (**H**) Shared genes between SFARI autism database and klf7 target genes in N2A cells. The number of ASD risk genes in the SFARI database bound by klf7 is given, and the total number in each subset is presented in parentheses. (**I**) Functional enrichment analysis of ASD risk genes targeted by klf7 in N2A cells. These ASD target genes were enriched for functions related to neurons and synapses. ASD, Autism spectrum disorder; klf7, Krüppel-like transcription factor 7; ODCs, oligodendroglial cells; log2FC, log2(Fold Change); SFARI, Simons Foundation Autism Research Initiative; CNS, central nervous system.

**Figure 2 ijms-23-03376-f002:**
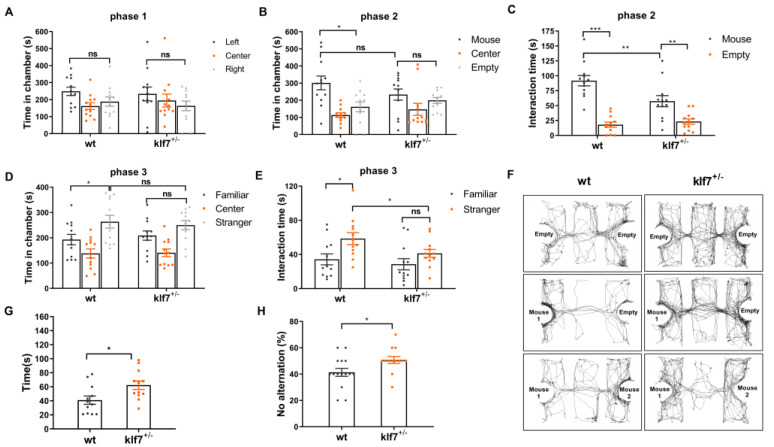
Loss of klf7 causes the mice to develop the core symptoms of ASD. (**A**–**F**) Three-chamber experiment. (**A**) All genotype mice had no preference for either of the two empty cages. (**B**) Whereas WT mice spent more time in the stranger mouse-containing caged chamber than in the empty cage-containing chamber, klf7^+/−^ mice had no preference for either chamber. (**C**) Klf7^+/−^ mice interacted with the stranger mouse for a shorter amount of time than WT mice. (**D**) Whereas WT mice spent more time in the chamber containing the novel mouse than in the chamber containing the familiar mouse, klf7^+/−^ mice had no preference for either chamber. (**E**) Klf7^+/−^ mice interacted with the novel mouse for a shorter period of time than WT mice did. (**F**) Representative tracks show the time spent in different phase of three-chambers social interaction experiment for WT mice and klf7^+/−^ mice. (G-H) Klf7^+/−^ mice exhibited stereotypical behaviors. (**G**) In the self-grooming test, klf7^+/−^ mice spent more time in self-grooming than WT mice. (**H**) In the Y maze spontaneous selection experiment, klf7^+/−^ mice preferred the original target arm, while WT mice preferred to explore the novel arm. The result is indicated by the ratio of entries into the original target arm to the entries into the novel arm in 10 trials. The data are presented as the mean ± SEM. *n* = 12 for WT and klf7^+/−^ mice. * *p* < 0.05, ** *p* < 0.01, and *** *p* < 0.001 by two-way ANOVA and unpaired t test; klf7, Krüppel-like transcription factor 7; ASD, autism spectrum disorder; WT, wild type.

**Figure 3 ijms-23-03376-f003:**
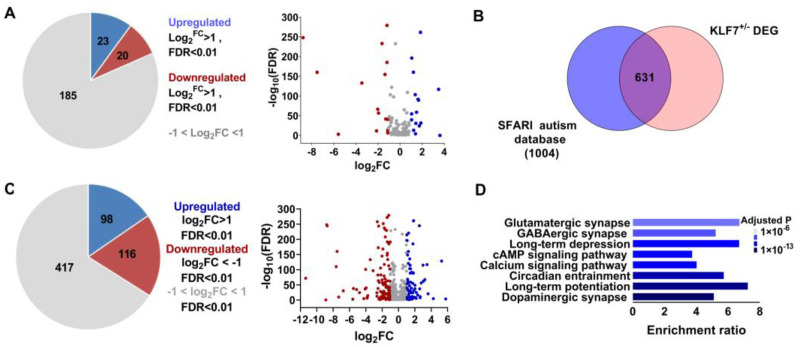
ASD risk genes are dysregulated in klf7^+/−^ mice. (**A**) Some klf7 target ASD risk genes were strongly dysregulated in klf7^+/−^ mice compared with WT mice. Of the 228 ASD risk genes targeted by klf7, the expression levels of 43 genes were significantly altered, with 23 being upregulated and 20 being downregulated (log2FC > 1 or log2FC < −1, false discovery rate (FDR) < 0.1). (**B**) A total of 631 genes were both identified as DEGs in klf7^+/−^ mice and found in the SFARI database. The number of ASD risk genes in the SFARI database disrupted by klf7 loss is indicated, and the total number in each subset is presented in parentheses. (**C**) Of the 631 autism genes in (**B**) that were disrupted by klf7 deletion, 98 were significantly upregulated (log2FC > 1), 116 were significantly downregulated (log2FC < −1), and 417 showed minimal changes (−1 < log2FC < 1). (**D**) The 631 autistic genes disrupted in klf7^+/−^ mice shown in (**B**) were functionally annotated. *n* = 4 for A; klf7, Krüppel-like transcription factor 7; ASD, autism spectrum disorder; DEGs, differentially expressed gene; SFARI, Simons Foundation Autism Research Initiative; log2FC, log2(Fold Change); TFs, transcription factors.

**Figure 4 ijms-23-03376-f004:**
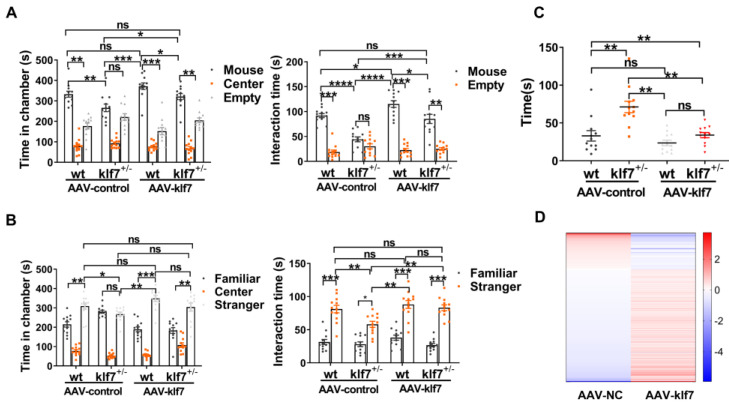
Increasing klf7 levels rescue the core symptoms of ASD in klf7^+/−^ mice. (**A**,**B**) Histogram showing the amount of time spent in the chamber containing either the stranger mouse or the novel mouse and the amount of time spent interacting with the stranger mouse or novel mouse by AAV-control-injected WT mice, AAV-control-injected klf7^+/−^ mice, AAV-klf7-injected WT mice and AAV- klf7-injected klf7^+/−^ mice in the three-chambers sociability test. Whereas klf7^+/−^ mice in the AAV-control group did not show an obvious social preference, klf7^+/−^ mice in the AAV-klf7 group showed improved social ability. Notably, elevating klf7 levels rescued the abnormal social ability of klf7^+/−^ mice. (**C**) Histogram showing the amount of time the AAV control-injected WT mice, AAV control-injected klf7^+/−^ mice, AAV- klf7-injected WT mice and AAV-klf7-injected klf7^+/−^ mice spent grooming in the 10 min self-grooming test. Klf7^+/−^ mice in the AAV-control group displayed an increased grooming time, indicating repetitive behavior. The abnormal grooming behavior was alleviated in the AAV-klf7 group. (**D**) Summary graph showing autism-related genes dysregulated after klf7 deletion whose expression was restored by elevating klf7 levels in klf7^+/−^ mice. The data are the mean ± SEM. *n* = 12 per group in A-C. * *p* < 0.05, ** *p* < 0.01, *** *p* < 0.001, **** *p* < 0.0001, ns: no significance by one-way ANOVA and two-way ANOVA; klf7, Krüppel-like transcription factor 7; ASD, autism spectrum disorder; AAV, adeno-associated virus; WT, wild type.

**Figure 5 ijms-23-03376-f005:**
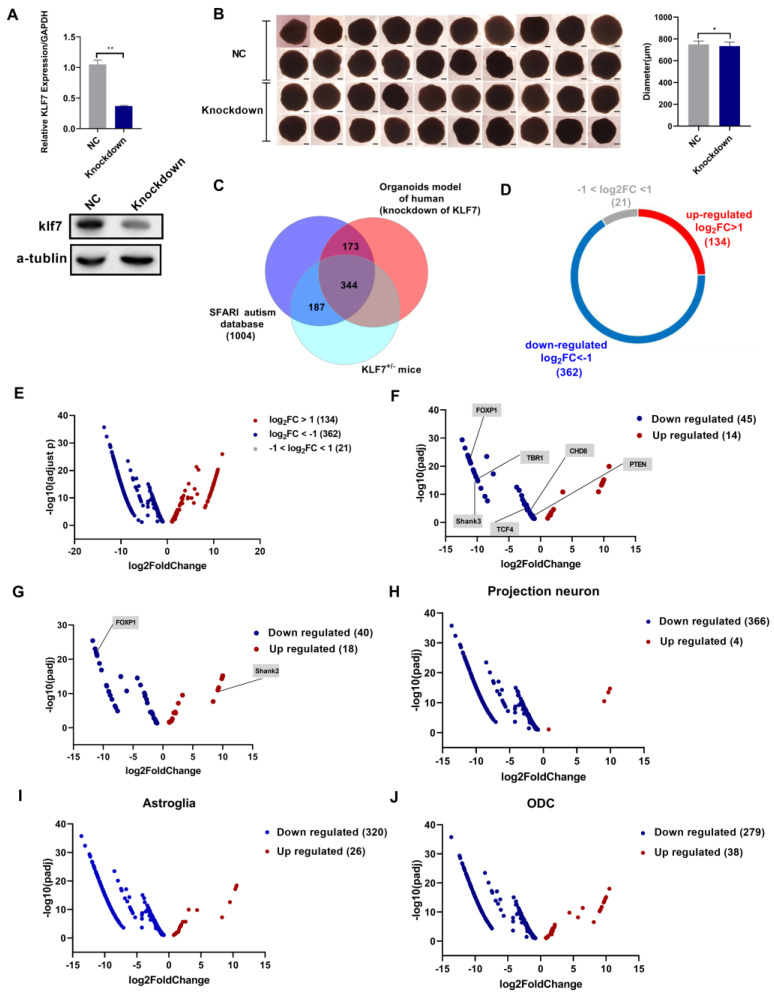
Klf7 knockdown in a human induced pluripotent stem cell (hiPSC)-derived organoid model. (**A**) Klf7 knockdown efficiency measured by qPCR (*n* = 3) and Western blot (*n* = 3) in organoids (90 days after differentiation). (**B**) The morphology of organoids in the control group and klf7 knockdown group after 90 days of differentiation. Klf7 knockdown significantly decreased the diameter of organoids; the average diameter of the control group was 750.8 μm, and the average diameter of the klf7 knockdown group was 734.8 μm. *n* = 20. Scale bar = 500 μm. (**C**) Genes shared between the SFARI autism database, klf7^+/−^ mice and klf7 knockdown organoids. The total number in each subset is noted in parentheses. (**D**,**E**) Of the 517 ASD risk genes that were dysregulated after klf7 deletion in hiPSC-derived organoids, 134 showed significantly upregulated expression (log2FC > 1), 362 showed significantly downregulated expression (log2FC < −1), and 21 showed minimal changes (−1 < log2FC < 1). (**F**) ASD risk genes (including 14 that showed upregulated expression and 45 that showed downregulated expression, adjusted *p* < 0.01) found to be dysregulated in klf7 knockdown organoids and identified as high-confidence genes in the ASD population by exome sequencing are indicated in the volcano plot. Some high-confidence ASD risk genes are shown. (**G**) ASD risk genes (including 18 that showed upregulated expression and 40 that showed downregulated expression, adjusted *p* < 0.01) dysregulated in klf7 knockdown organoids and identified as high-confidence genes by Willsey et al. are indicated in the volcano plot. Some high-confidence ASD risk genes are shown. (**H**–**J**) The plot shows ASD-related DEGs in klf7 knockdown organoids whose expression patterns were similar to those the brains of ASD patients. The data are presented as the mean ± SEM. *n* = 12 for A. * *p* < 0.05, ** *p* < 0.01 by unpaired t test. hiPSCs, human induced pluripotent stem cells; klf7, Krüppel-like transcription factor 7; qPCR, quantitative polymerase chain reaction; ASD, autism spectrum disorder; SFARI, Simons Foundation Autism Research Initiative; log2FC, log2(Fold Change); DEGs, differentially expressed genes.

## Data Availability

ChIP-seq of GFP-tagged klf7 in HEK293 cells are available in GEO under accession numbers GSM2026749. RNA-seq data and ChIP-seq data generated in this study have been submitted to https://zenodo.org/. DOI number for RNA-seq data of 1-month-old wild-type mice and 1-month-old klf7^+/−^ mice is 5236120 (https://zenodo.org/search?page=1&size=20&q=5236120, accessed on 23 August 2021). DOI number for RNA-seq data of WT/AAV-control mice, klf7^+/−^/AAV-control mice, WT/AAV-klf7 mice and klf7^+/−^/AAV-klf7 mice is 5242635 (https://zenodo.org/search?page=1&size=20&q=5242635, accessed on 24 August 2021). DOI number for RNA-seq data of klf7 knockdown human brain organoid model is 5242821 (https://zenodo.org/search?page=1&size=20&q=5242821, accessed on 24 August 2021). DOI number for ChIP-seq data of HA-tagged klf7 in N2A cells is 5243430 (https://zenodo.org/search?page=1&size=20&q=5243430, accessed on 24 August 2021).

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
