# Peer review of "Krüppel-like Transcription Factor 7 Is a Causal Gene in Autism Development"

_ijms, 2022, doi:10.3390/ijms23063376_

Round 1

Reviewer 1 Report

The authors made a complete and well done work on klf7; I think it's a paper of great importance in the field, with an amazing experimental section and with relevance also for clinicians. I've no suggestions. 

Author Response

We thank the reviewer for reading our paper carefully and giving the above positive comments. Thank you very much!

Reviewer 2 Report

===========

-----------------

Manuscript: ijms-1607857

Krüppel-like transcription factor 7 is a core transcriptional regulator in autism development,” by Hui Tian, et al.

-----------------------------------------------------------------

The manuscript presents results about effects of klf7 on expression of genes related to ASD (autism spectrum disorder). Such types of studies are important because they can provide further insight into genetic regulation of ASD and contribute to the development of appropriate therapeutic interventions. The authors tried to solve one of the main challenges related to ASD and identify a core set of ASD related genes. They tried to show (1) “that klf7 is one of the master regulator genes in ASD,” and (2) changes of its expression levels can reverse ASD and/or ASD symptoms. Unfortunately, some of these claims are not supported by the presented evidence, and statements made by the authors can lead to false or biased conclusions.

The manuscript can be divided into two parts: (1) results reported by the authors in this manuscript, (2) comparison of their results with available knowledge. Some of the authors’ statements are not supported by significant results, but all results (significant and non-significant) were included in the comparison analyses. I would suggest either to separate the manuscript into two parts or to delete non-significant results, redoing analyses by excluding those non-significant results and carefully rewriting claims and conclusions based on presented evidence.

I would like to highlight additional issues as described below. I hope that those comments and suggestion can help to improve the quality of the manuscript and made results more transparent and conclusions clearer.

(1) The authors made contradictory statements and considerations, which demonstrate that the claims and conclusions are not supported by the presented evidence.

Lines 398-401: “Klf7-deficient mice showed the core symptoms of ASD and could not distinguish the target quadrant in the water maze experiment, suggesting that klf7-deficient mice had impaired spatial learning ability and memory, which is consistent with the cognitive delay and intellectual disability seen in patients with klf7 mutations[17].”

Lines 406-407: “Our klf7+/- model mice recapitulate the symptoms of ASD and can function as models for studying the pathogenesis of ASD.”

One of the claims reported by the authors in the abstract is not supported by the presented evidence, namely (lines 20-21): “When klf7 levels were increased in the central nervous system (CNS) in klf7+/- adult mice, behavioral abnormalities and ASD gene dysregulation were rescued in the adults, suggesting transcriptional regulation.”

The latter claim contradicts to what the authors wrote about the results of their study, which is not reported in this manuscript (lines 297-299): “However, improving KLF7 expression did not significantly improve the mice's performance in water mazes, nor did they improve spatial learning and memory (the data were not shown).”

If “improving KLF7 expression did not significantly improve the mice's performance in water mazes,” maybe, the KLF7 or its expression levels is not directly related to the core symptoms of ASD and, maybe, there is something else. Could the authors explain how “klf7+/- model mice recapitulate the symptoms of ASD” if “improving KLF7 expression did not significantly improve the mice's performance in water mazes, nor did they improve spatial learning and memory (the data were not shown)”, while the latter one is related to ASD symptoms? Could the authors comment on the discrepancy in their statements and results/conclusions/suggestions?

(2) Lines 271-275: “… This indicates that mutations in these genes [probably, the authors refer to genes in Supplemental_Fig_S5E] mediate the development of autism and are not entirely dependent on klf7 regulation, suggesting that the phenotype of klf7-deficient mice may be mediated by the cumulative or synergistic effects of multiple ASD genes in the transcriptional network, further highlighting klf7 is a master regulator in ASD.” The mentioned independent regulation can be related to independent mechanisms. Therefore, the conclusion that this independence is “highlighting klf7 is [as] a master regulator in ASD” can be incorrect or misleading.

(3) Figure 4. The quality of this figure did not allow readers to make conclusions about significance of the presented results. I would propose to improve the quality of this and other figures making their elements (which are relevant to the main text) visible and readable. Moreover, it seems that the conditions were different in different experiments, for instance, measurement time. If the experimental conditions were changed these changes should be clearly described in the Methods section and in the main text. Some of the data do not sum up to the numbers described in Methods. Some of the results (rightmost columns for klf7+/- mice) do not demonstrate significant differences between the groups. I would suggest improving quality of representation in Figure 4 and more clearly describe whether the difference in the results was significant or not in each of the tests performed/compared, otherwise, I would recommend removing subsection 2.6 and Fig 4 from the manuscript.

(4) In the main text and in Supplemental Materials and Methods, “Data availability” section, the authors wrote that they made data publicly available for “RNA-seq data of 1 month old [1] wild type mice and [2] Klf7+/- mice, RNA-seq data of [3] AAV-mediated klf7 overexpression mice” It seems that something is missing, because in Supplemental Table S3 the authors presented results for (A) AAV-KLF7 group, and (B) AAV-NC group, which can be derived from data for four groups but not three as described in the “Data availability” section.

(5) Lines 301-302: “The expression of 427 of the 634 ASD risk genes was reversed (Figure 4D and Supplemental_Table_S3),…” From 634 genes presented in Supplemental Table S3, 15 and 11 genes demonstrated significant differences of the expression in AAV-KLF7 and AAV-NC group, respectively. Therefore, Fig4(E) is not relevant to this study but to a KNOWN SET of genes related to ASD and reported in the literature. Finally, Lines 301-308 and the conclusion made (Lines 307-308), “These results further demonstrate that klf7 is an up-stream regulator of ASD genes in the nervous system,” are not supported by the presented evidence.

(6) At the end of the Discussion section, the authors made a conclusion of their work (lines 448-451): “Due to klf7 overexpression promotes axonal regeneration after nerve injury and leads to functional recovery [55-57], we increase the level of klf7 not only alleviated the core symptoms of ASD but also reversed the abnormal expression of ASD genes.” This conclusion is not supported by the presented evidence, see (1-5).

(7) Lines 38-40: The following statements made by the authors look contradictory.

“To date, more than 1,000 genes have been shown to be associated with ASD [6], but none of these genes account for more than 1% of autism cases [7]. Therefore, it is crucial to identify which of genes contribute to ASD. Scientists are working to identify core regulatory genes, which remains a large challenge [8].”

If numerous genes (more than 1000 genes) contribute to ASD then a set of ASD genes will be affected by ASD definition. On the other side if ASD is a continuous spectrum of conditions it is possible that there are no core regulatory genes that contribute to ALL parts of the spectrum.

Could the authors express their point of view more clearly?

(8) Lines 35-38: Could the authors rewrite the following sentence making it clearer?

“A minority of ASD cases are syndromic forms caused by single gene mutations; in contrast, the vast majority of cases are complex, idiopathic ASD that involve variants in hundreds of genes that have small effects [5].”

(9) Lines 251-252. The authors wrote: “The regulatory network was reconstructed with dysregulated ASD genes and genes with klf7-binding sites in the promoter.” From this sentence it follows that the authors find one and only one, i.e. unique, combination of the regulatory network related to ASD. Could the authors comment on this? I would suggest rewriting this sentence by changing emphasis from the word “reconstructed” to something more realistic and related to the present study, “build” or something similar.

Minor corrections:

(10) Could the authors introduce notations to abbreviations where they are used for the first time? For instance, what log2FC stays for (see line 61)?

(11) Lines 64-65: Could the authors clarify what does the following sentence mean?

“Klf7 and ASD risk genes share the same spatial expression period.”

(12) In the Supplemental Methods section, subsection “Open field test”, there are no units provided: “The open field apparatus was a 40 × 40 × 30 box.”

(13) In the Supplemental Methods section the authors wrote: “We used clusterProfiler R package (3.8.1) and clusterProfiler R package (3.8.1) to test the statistical enrichment of differential expression genes in KEGG pathways and Gene Ontology (GO) enrichment.” It seems that they are mentioned the same R package, clusterProfiler R package (3.8.1), twice. Could the correct the text?

(14) In the “Data availability” sections, see main text and Supplemental Materials and Methods, the authors wrote: ” RNA-seq data of 1 month old wild type mice and Klf7+/- mice, RNA-seq data of AAV-mediated klf7 overexpression mice, RNA-seq data of klf7 knockdown human brain organoid model, and ChIP-seq data of HA-tagged klf7 in N2A cells generated in this study have been submitted to web site at https://zenodo.org/  https://zenodo.org/ with DOI number 10.5281/zenodo.5236120、10.5281/zenodo.5242635、10.5281/zenodo.5242821 and 10.5281/zenodo.5243430”

Could the authors provide proper references?

(15) I would suggest increasing font size in Figures 1(A)-A(C) in order to improve readability.

(16) In the main text, I would suggest providing references to respective subsections of the Supplemental Methods where it is appropriate and relevant.

(17) Lines 112-113: The authors wrote: “More than 90% of klf7-/- mice died during the first two months after birth (Supplemental_Fig_S2C).” Could the authors provide additional information in the text about the cause(s) of death in klf7-/- mice.

(18) In the main text, could the authors provide/report information about sample size where it is appropriate?

(19) It is difficult to resolve details in Figure 3. Could the authors improve its readability by changing font size and shape/size of symbols used?

(20) Lines 208-212: Probably the following sentence should read as follows: “631 genes were shared with SFARI database (Figure 3B and Supplemental_Table_S2); 215 genes of these genes [demonstrated] prominent changes (including 98 showing upregulated expression and 116 showing downregulated expression, log2FC >1 or log2FC<-1, false discovery rate [FDR]<0.01) and the remaining 417 genes showed small changes (Figure 3C, -1<log2FC<1, false discovery rate [FDR]<0.01).” Could the authors explain what does “prominent changes” mean in statistics?

(21) Supplemental_Fig_S5. There is no notation to (E) subfigure.

(22) Supplemental_Fig_S5. The last sentence in the figure caption reads: “P*<0.05, P**<0.01 by unpaired t test in E. n=9 for E.” Could the authors correct this sentence?

(23) Supplemental_Fig_S5E. This figure demonstrates non-significant difference in the expression of AUTS2, CNTNAP2. Why the authors wrote that this subfigure presents “Validation of the expression of high confidence ASD risk genes by qRT-PCR” if some results are non-significant?

(24) Lines 252-253. “Genes are shown as circles, and TFs are shown as squares.” Circles are visible in Fig3(D), but I was not able to see any square. Could the authors provide a correction either to description or in figure?

(25) Lines 253-257: “Upregulated genes are shown with a red border, and downregulated genes are shown with a green border. Klf7 directly regulates ASD risk genes shown in blue and non-ASD risk genes shown in pink. Klf7 indirectly regulates ASD genes shown in yellow.” Colors in Figure 3(D) do not correspond to the description. Could the authors change the description or the colors?

(26) Lines 256-258: “The lines reflect a regulatory relationship: red lines indicate direct regulation by klf7, gray lines represent indirect regulation by klf7 and black dotted lines with red arrows are interactions predicted by database.” It is almost impossible to distinguish the lines as described. Could the authors fix this issue?

(27) Figure 4 suffers the same issues as described above.

(28) I would suggest checking the titles of Supplemental Tables S4A and S4B for misprints.

(29) Lines355-356: “Among 264 ASD genes targeted by human klf7, 38 were up-regulated and 97 were down-regulated (adjusted p<0.01, data were not shown).” I would suggest either to remove this sentence or to present supporting evidence.

(30) There are no explanations for abbreviations in Tables and Figures neither in the main text nor in the Supplemental Materials and Methods.

(31) Lines 404-406: “Our results also showed that insertion of the loxP site had no effect on mouse behavior (data were not shown).” I would suggest to delete statements that are not supported by the presented evidence.

(32) Lines 434-435: “In another ASD model of maternal immune activation (MIA), we found that klf7 was upregulated at E17.5 and down-regulated at P0[49].” Could the authors explain how they participated in a previously published study [49]?

(33) Could the authors spell out E17.5 and P0? Is it related to mice or other organisms?

(34) Could the authors explain/comment on why they repeat the Materials and Methods section in Supplemental Materials and Methods?

Reviewer 3 Report

In their study conducted on a murine model and supplemented with bioinformatic analysis, Tian et al. convincingly demonstrated, that Klf7 is a gene invlolved in autism development. Heterozygous klf7+/- mice showed autistic traits, moreover, they could be partially rescued after the transcription factor level had been restored. Besides, the bioinformatic assay showed that Klf7 deficiency had led to changes, though small, in the expression levels of as many as 419 ASD risk-associated genes. Taken together, these findings suggest that a heterozygous mutation in Klf may cause ASD in children, thus corroborating an early case report on four unrelated children with de novo mutations in Klf7 accompanied by low-functioning ASD, as well as few reports describing a patient exhibiting ASD traits and carrying a deletion in 2q33.3q34, where Klf7 is located.

However, in my opinion, the authors overhastily interpreted their valuable data as a discovery of the clue to most ASD cases, as I could understand by epithets ‘core’, ‘critical’, ‘master’. The bulk of available evidence is insufficient to regard the partial loss of this transcription factor as a major or frequent cause of ASD in human populations. There is still no data on the frequency of ‘broken’ Klf7 alleles in the population. A hypothesis of the critical role of one autosomal gene (e.g. Klf7) would fail to explain the distorted sex ratio in ASD with the bias toward male patients, and the rapid growth of autism prevalence. In few clinical cases so far published, severe impairments in addition to autistic features, such as microcephaly, dysmorphic features, hypotonia, feeding and swallow problems, and neuromuscular complications were reported in patients with a defect or deleted copy of Klf7. These signs are not characteristic for the majority of ASD cases. In the study, some klf7+/- mice and all klf7-/- mice exhibited decreased body weight and brain weight, consistent with microcephaly reported in human patients. Contrary to this, many ASD patients demonstrate macrocephaly, though microcephaly and normal head size are also observed. The loss of function is lethal in homozygous state (see line 46) and dominant in heterozygous state (haploinsufficiency) with serious consequences. So, there is no reason to conclude that this mutation can account for a condition with a prevalence of ~1% children, such as ASD in the USA. I recommend to write that, to date, we may only declare confirmation of discovering just one more ASD-associated gene, while its relevancy and real contribution to the ASD epidemiology will be a subject of future research.

For this reason, and guided by a desire to improve some other weak points, I’d like to suggest the following minor revisions to the text of manuscript:

  1. Lines 2 and 214: delete ‘core
  2. Line 12: replace ‘neurological’ with ‘neurodevelopmental’ or ‘psychiatric’
  3. Lines 14 and 39: check stating mere 1%; as I know, some syndromal ASD genes (Fmrp, Tsc1&2) can account for more than 1% of cases
  4. Lime 14: delete ‘So it is important to find the core gene that contribute to ASD’. [Most scientists now converge on idea that there is no single ‘core’ gene in ASD. And this phrase misleads to an impression that the study tried to find and even were successful in finding such a magic ‘master gene’]
  5. Lines 24-26: re-write ‘Conclusions’ as follows or in a similar way: ‘Conclusions: Our findings highlight a klf7 regulated network of ASD genes and provide new insights into the pathogenesis of ASD and promising targets for further research on mechanisms and treatments.’
  6. Line 28: delete ‘critical
  7. Line 44, 45, 46, 47: provide more details on normal Klf7 functions, if available
  8. Line 51: numerals 1 to 9 should be written in words (‘four unrelated individuals…’), this is a universal rule of scientific style
  9. Line 54: replace ‘critical’ with ‘causal’
  10. Line 81: change ‘consist with’ for ‘consistent with’
  11. Line 275: delete the whole line and put a dot in the end of line 274 after the word ‘network.’

Round 2

Reviewer 2 Report

The authors revised the manuscript substantially by addressing a major part of Reviewers’ comments and suggestions.

Unfortunately, some of the comments, questions and issues remained unanswered or ignored. In my review, comments (5) and (6) emphasized a problem/issue.

(2.1) The problem can be described as follows: when building a network of ASD genes related to klf7 (Figure 4(E)), the authors used ALL ASD related genes including genes with non-significant changes in their expression levels. By using this approach, the authors exaggerated significance of their results, which can mislead the interpretation of the findings reported in the manuscript.

(2.2) After the authors provided answers to a few of my questions, I can see that Figure 3(D) suffers the same issue as described in (2.1), i.e. the authors used non-significant results to exaggerate their findings and demonstrate that klf7 is related to the dysregulation of ALL known genes related to ASD, i.e. to the entire network of ASD related genes.

(2.3) I would like to provide an example of a hypothetical study which would use the same approach as the authors did in the presented study. Let us consider a network of genes related to lipid metabolism and select one gene, G1, from this network, which mediates some of the lipid related processes. In knockout mice the expression levels of other genes in the network will be altered, which will be caused by adaptations. One set of them will demonstrate an increase, but another set will demonstrate a decrease of the expression level. Some of these changes will be significant, but numerous will be non-significant (which would provide results like that presented by the authors in Figures 3(A) and 3(C)). However, if ALL genes (with significant and non-significant changes) will be included when “building” a network of genes related to the G1 gene (presentation that would be similar to Figure 3(D)), one can conclude that this gene, G1, is a “core”, i.e. a causal gene, in the network of lipid metabolism, which is not true.

(2.4) In Figures 3(D) and 4(E) and related conclusions, instead of presenting a network of genes with significant connections to klf7, the authors used ALL ASD related genes (with significant and non-significant relationships to klf7). This approach exaggerated significance of their results and can lead to incorrect conclusions.

(2.5) I would propose to delete Figures 3(D) and 4(E), and related conclusions OR, alternatively, to rebuild these figures by using significant results (connections to klf7) ONLY and reformulate related conclusions.

(2.6) The manuscript can be divided into two parts: (1) results reported by the authors in this manuscript, (2) comparison of their results with available knowledge, i.e. building a network of ASD related genes that are connected to klf7. In the revised version the authors corrected issues that were related to the first part, but they ignored to correct the issues described above (2.1-2.5), which are related to the second part.

(2.7) I would suggest either (A) to separate the manuscript into two parts as described above and publish respective results in separate papers (the first part was corrected, but the second part suffers of issues mentioned above); or (B) to delete non-significant results, rebuilding network of ASD genes SIGNIFICANTLY connected/related to klf7, redoing analyses by excluding those non-significant results and carefully rewriting claims and conclusions based on acquired evidence.

Round 3

Reviewer 2 Report

The authors revised the manuscript considering all suggestions and removed some materials that were not supported by the results. In my opinion, the message of this manuscript became clear and claimed findings are supported by presented evidence.